# Fatty Acid Amide Hydrolase (FAAH) Inhibition Modulates Amyloid-Beta-Induced Microglia Polarization

**DOI:** 10.3390/ijms22147711

**Published:** 2021-07-19

**Authors:** Maddalena Grieco, Maria Giovanna De Caris, Elisa Maggi, Federica Armeli, Roberto Coccurello, Tiziana Bisogno, Maria D’Erme, Mauro Maccarrone, Patrizia Mancini, Rita Businaro

**Affiliations:** 1Department of Biochemical Sciences, Sapienza University of Rome, Piazzale Aldo Moro 5, 00185 Rome, Italy; maddalena.grieco@uniroma1.it (M.G.); maria.derme@uniroma1.it (M.D.); 2Department of Experimental Medicine, Sapienza University of Rome, Viale Regina Elena 324, 00161 Rome, Italy; mariagiovanna.decaris@uniroma1.it (M.G.D.C.); patrizia.mancini@uniroma1.it (P.M.); 3Department of Medico-Surgical Sciences and Biotechnologies, Sapienza University of Rome, Corso della Repubblica 79, 04100 Latina, Italy; elisa.maggi@uniroma1.it (E.M.); federica.armeli@uniroma1.it (F.A.); 4Fondazione Santa Lucia IRCCS, Preclinical Neuroscience, Via del Fosso di Fiorano 64, 00143 Rome, Italy; roberto.coccurello@cnr.it; 5Institute for Complex Systems (ISC), C.N.R., Via dei Taurini 19, 00185 Rome, Italy; 6Endocannabinoid Research Group, Institute of Translational Pharmacology, CNR, Via Fosso del Cavaliere 100, 00133 Rome, Italy; 7Department of Biotechnological and Applied Clinical Sciences, University of L’Aquila, Via Vetoio snc, 67100 L’Aquila, Italy; mauro.maccarrone@univaq.it; 8European Center for Brain Research/Santa Lucia Foundation IRCCS, Via del Fosso di Fiorano 64, 00143 Rome, Italy

**Keywords:** URB597, *N*-arachidonoylethanolamine (AEA), fatty acid amide hydrolase (FAAH), microglia, actin cytoskeleton, cell migration, M1/M2 phenotypes, neuroinflammation

## Abstract

The ability of endocannabinoid (eCB) to change functional microglial phenotype can be explored as a possible target for therapeutic intervention. Since the inhibition of fatty acid amide hydrolase (FAAH), the main catabolic enzyme of anandamide (AEA), may provide beneficial effects in mice model of Alzheimer’s disease (AD)-like pathology, we aimed at determining whether the FAAH inhibitor URB597 might target microglia polarization and alter the cytoskeleton reorganization induced by the amyloid-β peptide (Aβ). The morphological evaluation showed that Aβ treatment increased the surface area of BV-2 cells, which acquired a flat and polygonal morphology. URB597 treatment partially rescued the control phenotype of BV-2 cells when co-incubated with Aβ. Moreover, URB597 reduced both the increase of Rho protein activation in Aβ-treated BV-2 cells and the Aβ-induced migration of BV-2 cells, while an increase of Cdc42 protein activation was observed in all samples. URB597 also increased the number of BV-2 cells involved in phagocytosis. URB597 treatment induced the polarization of microglial cells towards an anti-inflammatory phenotype, as demonstrated by the decreased expression of iNOS and pro-inflammatory cytokines along with the parallel increase of Arg-1 and anti-inflammatory cytokines. Taken together, these data suggest that FAAH inhibition promotes cytoskeleton reorganization, regulates phagocytosis and cell migration processes, thus driving microglial polarization towards an anti-inflammatory phenotype.

## 1. Introduction

Alzheimer’s disease (AD) is a neurodegenerative disorder characterized by the progressive and inescapable cognitive deterioration, whose main neuropathological hallmarks are the formation of intracellular neurofibrillary tangles and extracellular senile plaques, producing synaptic dysfunction, and ultimately, neural death. While neurofibrillary tangles are composed of hyperphosphorylated tau protein, senile plaques are associated with neurotoxicity and are formed by aggregates of the β-amyloid peptide (Aβ), stemming from the misprocessing of the Amyloid Precursor Protein (APP) [1,2,3]. To date, the molecular mechanisms underlying AD are not completely understood. McGeer and Eikelenboom (1994) were the first to clearly identify the inflammatory component in AD, as recently confirmed by Shippy et al. (2020) [4,5,6]. Indeed, several authors pointed out the ability of Aβ aggregates to activate the microglial cells, thus inducing the release of inflammatory mediators such as ROS, nitric oxide, and interleukins, all responsible for neuronal death [7,8,9,10]. Moreover, the postmortem examination of Alzheimer’s patients has confirmed the incidence of high levels of pro-inflammatory cytokines, chemokines, and inflammation mediators [11].

Microglial cells play a central role in maintaining brain homeostasis and are involved in resolving inflammation from trauma or infectious microorganisms by means of phagocytosis and/or anti-inflammatory mediators. However, while the initial inflammatory response can be neuroprotective for the brain, the persistence of a challenging stimulation and the resultant strong microglia activation may turn the first line of defense (e.g., released cytokines) into detrimental and auto-toxic reactions that lead to synaptic dysfunction and neuronal cell death.

To date, no drugs used in anti-AD therapy seem to improve the prognosis. Indeed, the current therapeutic strategies are aimed at symptoms alleviation or the slowing down of disease progression. Hence, there is the need to introduce innovative drugs capable of interfering with the pathophysiological mechanisms underlying AD. In this scenario, growing interest has been focused on the endocannabinoid (eCB) system that is viewed as a pro-homeostatic and pleiotropic signaling system activated as an adaptive response to multiple pathological conditions [12,13]. The eCB system consists of cannabinoid type-1 (CB_1_) and type-2 (CB_2_) receptors, endogenous lipid ligands such as *N*-arachidonoylethanolamine (AEA) and 2-Arachidonoyglicerol (2-AG), as well as proteins and enzymes involved in their biosynthesis and inactivation. The eCB system is also considered as part of a mechanism that may operate morphological [14], phenotypic, and functional changes of microglia and counteract the neuroinflammatory processes occurring in neurodegenerative diseases [15,16].

Studies performed both in vitro and in vivo have highlighted CB_1-2_ receptor-mediated inhibition of Aβ-induced neurotoxicity, gliosis, neuroinflammation, and memory deficits [17,18,19]. Moreover, the pharmacological activation of CB_1_ and/or CB_2_ receptors improves memory and/or cognitive impairments in both AD-like transgenic and pharmacological (e.g., via intracerebral Aβ infusion) mice models [18,20,21]. In addition, the increased expression of CB_2_ levels in astrocytes- and microglia-associated neuritic plaques strongly suggest a CB_2_-mediated neuroprotective effect via the stimulation of microglial proliferation and migration [18,22].

In the past few years, several lines of evidence have suggested that eCB degradative enzymes could be a target for the development of anti-AD therapeutic options. Both the expression and activity of fatty acid amide hydrolase (FAAH), the enzyme responsible for AEA degradation, are increased in astrocytes and microglia-associated neuritic plaque from post-mortem AD patients’ brains [17]. A reduction of AEA levels was also reported in the midfrontal and temporal cortex, where AEA levels inversely correlated with Aβ_1-42_ content [23]. Furthermore, the pharmacological modulation or genetic ablation of FAAH exerted different beneficial effects in terms of Aβ accumulation, neuroinflammation, and cognitive decline [24,25,26]. Recently, Tanaka et al., (2019) reported that the pharmacological inhibition or ablation of FAAH in BV-2 microglia cells reduced the LPS-induced release of inflammatory mediators and pro-inflammatory cytokines, and siRNA FAAH BV-2-transfected cells led to the reduced expression of pro-inflammatory genes. These findings suggest that genetic suppression and/or pharmacological inhibition of FAAH might modulate microglial phenotypes and produce anti-inflammatory effects via different mechanisms [27].

Taken together, these studies prompted us to investigate the effects produced by FAAH inactivation on microglial inflammatory processes. From among the different FAAH inhibitors developed so far, we selected the 3-(3-carbamoylphenyl)-phenyl N-cyclohexylcarbamate (URB597), as it is one of the best known compounds with a high affinity to FAAH. Hence, we assessed URB597 efficacy in terms of its ability to modulate morphological and functional changes in Aβ_25–35_-induced microglial activation in BV-2 cells. Our results show that URB597 treatment exerts a potent anti-inflammatory action by inhibiting BV-2 cell microglial polarization, by promoting cytoskeleton reorganization via cell migration and phagocytosis processes as well as by modulating the expression of pro- and anti-inflammatory markers.

## 2. Results

### 2.1. Aβ_25–35_ Induces Upregulation of the Iba1 Microglia Marker

Since Iba1 is a selective marker of microglial activation following nerve injury, central nervous system ischemia, inflammatory conditions, and several other forms of brain damage, we assessed whether Aβ_25–35_ could induce microglia activation by immunofluorescence and western blot analyses [28,29]. We observed an upregulation of Iba1 in BV-2 cells treated with 30 µM of Aβ_25–35_ for 24 h, thus confirming microglial activation. URB is able to decrease Aβ_25–35_-induced microglia overactivation (Figure 1).

### 2.2. URB597 on BV-2 Cell Viability

To determine the effect of Aβ_25–35_ in the presence or absence of URB597 on BV-2 cell viability, the Trypan blue assay and MTT analysis were performed. We selected 5 µM of URB597 concentration based on the dose-response curves on BV-2 cells (data not shown). Microglia cells were pre-treated with 5 μM URB597 and incubated with 30 μM Aβ_25–35_ for 24 h. Figure 2A shows a significant increase of cell death in the presence of Aβ_25–35_ as compared to control cells, whereas URB597 reversed Aβ_25–35_-induced cell death. URB597 alone had no effect.

In addition, we performed an MTT assay using cells treated as described above. The data obtained show that Aβ_25–35_ induced a decrease in cell viability by around 40% at 24 h, while a slight proliferative effect of Aβ_25–35_ was observed at 48 h. Treatment with URB597 alone did not interfere with cell survival and did not prevent Aβ_25–35_ challenge at the mitochondrial level (Figure 2B).

### 2.3. Aβ_25–35_ and URB597 Effects on FAAH Enzyme Activity

Since FAAH is a selective eCB-degrading enzyme, we investigated whether Aβ_25–35_ and URB597 might modulate its activity. For this purpose, BV-2 cells were pre-treated with 5 μM URB597 for 4 h and incubated with 30 μM Aβ_25–35_ for 24 h. Notably, Aβ_25–35_ induces neurotoxicity and neuroinflammation, comparable to a full-length Aβ_1-42_ peptide [30].

Aβ_25–35_ did not have any significant effect on FAAH activity at 24 h (Figure 3). URB597 was also able to inhibit FAAH activity in the presence of Aβ_25–35_.

### 2.4. URB597 Reverts Morphological Changes Induced by Aβ_25–35_

Microglia rapidly responds to brain injury and disease by altering its morphology and adjusting its phenotype towards an activated state. In the activated state, microglia shifts from its resting form to an ameboid form (M1/M2) [31]. Cells increase their surface area and acquire a flat morphology. To evaluate the effects of FAAH inhibition on cell morphology, we performed an immunofluorescence analysis through the phalloidin staining of F-actin on BV-2 cells pre-treated with 5 μM of URB597 for 4 h and incubated with 30 μM of Aβ_25–35_ (Figure 4). Cell morphology was analyzed at different time points (1 h, 3 h, 6 h, 24 h).

The results show that control cells exhibited a very small cell body. They had long cell ramifications and actin was organized predominantly in the filopodia, used by the cells to explore the surrounding environment. At all time-points analyzed, cells treated with Aβ_25–35_ increased their surface area, acquiring a flat and polygonal morphology. Furthermore, cells retracted the branched processes that is typical of microglia in its resting form. On the other hand, cells treated with URB597 had a more rounded morphology, characterized by the shrinkage of the cellular body and the presence of several cellular processes. BV-2 cells co-treated with URB597 and Aβ_25–35,_ showed a phenotype similar to that obtained with only URB597 (Figure 4A). These results were also confirmed by the quantitative analysis carried out by measuring the cellular area expressed in μm^2^ at 24 h. The quantitative analysis confirmed that the treatment with URB597 alone produced surface areas similar to those of control cells, while after stimulation with Aβ_25–35_ the cells underwent an enlargement of their soma. The cell had an area of 440 μm^2^ while the control sample was at 340 μm^2^. In the combined treatment, URB597 reduced the area to values comparable to those of controls, indicating the capacity of URB597 to restore the amoeboid phenotype observed in the presence of Aβ_25–35_ (Figure 4B).

### 2.5. Effect of URB597 on Cellular Migration

One of the aspects of microglia activation is the acquisition of a migratory phenotype, which is essential for the cells to move towards the insult site. In this study, we performed a scratch assay in order to evaluate URB597-induced migratory capacity. The cells were pre-treated with 5 µM URB597 for 4 h in the presence or absence of 30 µM Aβ_25–35_ for 2 h, 4 h, 6 h, and 24 h. Migration was quantified by counting the cells that migrated from the border of the scratch to the uncovered areas, and the values obtained were reported on the graph (Figure 5B).

As shown in Figure 5, treatment with Aβ_25–35_ induced migration just 2 h after stimulation, as compared to the control cells. We demonstrated that URB597 alone did not affect cell migration, and that URB597 pre-treatment reduced BV-2 migration induced by Aβ_25–35._ The migratory effect was directly proportional to the number of incubation hours.

### 2.6. URB597 Increases Microglia Phagocytic Capacity

To determine whether URB597 has a role in phagocytosis, we assessed FITC-dextran microglial uptake. BV-2 cells pre-treated with 5 µM URB597 for 4 h and incubated with 30 µM Aβ_25–35_ for 24 h were incubated for 1 h with FITC-dextran and analyzed by a microscope; the fluorescence images are shown in Figure 6A. As shown in Figure 6B, we evaluated the intensity of the fluorescent dextran in the different treatments. BV-2 cells treated with Aβ_25–35_ showed a reduced ability to absorb dextran compared to untreated cells. The sample treated only with URB597 showed an intensity value comparable to that obtained by IL-10 stimulation. In the combined treatment with Aβ_25–35_ and URB597, the inhibitor turned out to be capable of increasing the intensity value as compared to the Aβ_25–35_ sample. LPS and IL-10 were used as reference compounds that could reduce [32] or increase phagocytosis [33,34]. Moreover, the number of dextran-positive cells was quantified on the total cells and the results were reported as percentage (Figure 6C). After stimulation with Aβ_25–35_ as well as with LPS, the number of dextran-positive cells was reduced by approximately 80% relative to the control cells. On the other hand, treatment with URB597 alone showed values similar as the control but lower if compared to the IL-10 treated sample. In the Aβ_25–35_ and URB597 combined treatment, the inhibitor improved the number of dextran-positive cells, with higher values of phagocytic cells as compared with the Aβ_25–35_ treated sample.

### 2.7. URB597 Affects Rho GTPases Activity

Cytoskeleton plasticity and the formation of actin-rich structures, with consequent morphological modifications, are regulated by the Rho GTPase protein family. Since the Rho GTPase family is a master regulator of cytoskeletal reorganization and plays an important role in membrane trafficking [35,36], we investigated whether the Rho and Cdc42, components of the Rho GTPase family, were able to regulate the different activated states of microglia cells. To evaluate the influence of the FAAH inhibitor on this class of proteins, we performed pull-down analyses on the total protein extraction from BV-2 cells pre-treated with 5 μM URB597 for 4 h and incubated with 30 μM Aβ_25–35_ for 24 h. We examined RhoA, which is involved in the assembly of contractile actin, and Cdc42 proteins that is involved in filopodia formation.

This analysis allows for the precipitation of active GTPases proteins through a specific binding protein. Our results show a marked increase of RhoA activity in the samples treated with Aβ_25–35_ as compared to control cells, whereas a decrease was observed in the URB597-treated samples with respect to Aβ_25–35_ samples (Figure 7A). As shown in Figure 7B, Cdc42 activity significantly increased in samples treated with URB597, considering both against Aβ_25–35_ and control cells samples.

### 2.8. URB597 Reduces mRNA IL-1β and TNF-α Expression, Increases TGF-β and IL-10 Expression

The activation of Rho GTPases supports the hypothesis of Aβ-driven cytoskeleton rearrangement, ultimately leading to an increase of migration and a reduction of phagocytosis. Since these effects were counteracted by URB597, we aimed at determining the level of inflammation in our BV-2 cell model.

The expression of pro-inflammatory cytokines such as IL-1β and TNF-α as well as anti-inflammatory cytokines such as TGF-β and IL-10, were analyzed by qPCR. BV-2 cells were pre-treated with 5 µM URB597 for 4 h and incubated with 30 µM Aβ_25–35_ for 1 h and 24 h. The results presented in Figure 8A,B demonstrate that IL-1βand TNF-α expressions increased within 1 h in BV-2 cells treated with Aβ_25–35_ as compared to the control. On the contrary, a reduction of cytokines was observed in the sample with the combined treatment as compared to Aβ_25–35_ alone. No changes were detected for IL-1β and TNF-α levels in BV-2 cells incubated with 30 µM Aβ_25–35_ for 24 h. We also observed that treatment with Aβ_25–35_ did not modify TGF-β expression, while URB597 alone or in combination with Aβ_25–35_ upregulated TGF-β at 24 h, as shown in Figure 8C.

Moreover, both Aβ_25–35_ and URB597 administration, alone or in combination, induced an increase of IL-10 expression within 1 h, while after 24 h of stimulation the effect was persistent only in the Aβ_25–35_ and URB597 combined treatment (Figure 8D).

### 2.9. URB597 Modulates Both iNOS and Arg-1 Expression

The activation of microglia is a polarized process that can lead either to the potentially neurotoxic M1-activated phenotype or to the neuroprotective M2-activated phenotype [37]. To evaluate the M1 or M2 states in the treated cells, we performed both immunofluorescence and RT-qPCR analyses and assessed the level of nitric oxide synthase (iNOS) and Arginase-1 (Arg-1), which are markers of the M1 and M2 microglia phenotypes, respectively (Figure 9). BV-2 cells were pre-treated with 5 µM URB597 for 4 h and incubated with 30 µM Aβ_25–35_ for 24 h. LPS and IL-4 were considered as positive controls for iNOS and Arg-1 markers, respectively. As shown in Figure 9A–D, the stimulation of BV-2 cells with Aβ_25–35_ induced a significant enhancement of iNOS, whereas a reduction of Arg-1 was observed in comparison to control. The combination of URB597 and Aβ_25–35_ induced a decrease in iNOS expression and an increase in Arg-1 with respect to Aβ_25–35_. URB597 alone did not affect the expression of both markers.

## 3. Discussion

Neuroinflammation, activated microglia, and abnormalities in cytoskeletal organization are a common feature of many neurodegenerative disorders, including AD [38,39]. Although microglia polarization and plasticity are adaptive changes aimed at the surveillance of the surrounding environment to identify possible harmful processes, the mechanism underlying this process is not yet completely understood [40,41].

Several studies demonstrated that eCBs act as immunomodulators by inhibiting cytokine and chemokine production and microglial cell activation [42,43]. To validate our experimental settings, we assessed Aβ_25–35_ induced microglia activation, which was partially counteracted by URB597, evaluating both the immunofluorescence signal of Iba1-positive cells and the Iba1 protein level, supporting the efficacy of URB597 on FAAH activity in BV-2 cells [27].

In this study, we investigated whether the inflammatory challenge induced by Aβ_25–35_ on microglia cells might be prevented by FAAH inhibition. Moreover, we reported no differences in FAAH activity in BV-2 cells treated with Aβ_25–35_ as compared to control cells.

However, it should be noted that while URB597 was able to counteract Aβ_25–35_ induced cell death, cell viability was not completely restored by the same treatment. One explanation for this potential discrepancy is that cell viability may depend on the MTT assay, which provides different measures of mitochondrial activity. Notably, by enhancing FAAH substrate levels, URB597 can modulate cell proliferation [44,45] and mitochondrial bioenergetics [46,47]. In particular, the mitochondrial cascade hypothesis supports the existence of a reinforcing cycle of mitochondrial dysfunction (including the impairment of oxidative phosphorylation, reactive oxygen species production, and the alteration of mitochondrial dynamics) that, together with Aβ formation, leads to AD pathogenesis and cognitive decline [48,49]. Recently, in an in vivo model of kainic acid-induced epilepticus status in rats, URB597 treatment was shown to revert the ultrastructural alterations in the ER, mitochondria, and hippocampal neurons in a way dependent on both degree of injury and URB597 treatment [50].

Thus, a different response in cell viability might be assessed in microglial cells by using a longer time window for URB597 incubation.

With these parameters established, the first point was focused on cell morphology, which revealed specific changes after Aβ_25–35_ stimulation—from the resting ramified form to an amoeboid shape, with an enlarged cell body and very few extensions. This shape, typical of activated pro-inflammatory state M1, supports cell migration. By contrast, URB597 was able to affect the rearrangement of morphology in Aβ_25–35_ challenged BV-2 cells and significantly decreased Aβ_25–35_ dependent migration. Moreover, our data indicate that URB597 also modulates microglia phagocytosis since it restored the phagocytic capacity of BV-2 cells that was reduced by the Aβ_25–35_ challenge.

URB597 is reported to have no activity on other cannabinoid-related targets as well as on other serine hydrolase superfamily enzymes, as assessed by the activity-based protein profiling (ABPP) approach [27,51]. By changing eCB levels, URB597 might indirectly activate CB_1_ and CB_2_ receptors and contribute to the regulation of microglial migration and phagocytic activity towards an anti-inflammatory microglial phenotype [52,53,54]. It should also be noted that the effects of FAAH inhibition might be mediated by other receptors such as PPAR-α or -γ and TRPV1 [55], or other FAAH substrates such as N-palmitoylethanolamine [53]. Moreover, effects of URB597 on the expression of tyrosine hydroxylase via CB_1_- and FAAH-independent mechanisms [56] or its interaction with different carboxypeptidases [57] have also been reported. Although our results add another piece of evidence in support of the therapeutic potential of FAAH inhibitors, caution is needed regarding the modulation of phagocytosis. Indeed, during early brain development, microglial phagocytosis mediates synapse removal as part of a physiological process required for proper brain maturation [58]. However, during aging or in several pathological conditions, the process of synaptic pruning becomes reactivated and may contribute to the cognitive decline observed in neurodegenerative diseases [59]. Although phagocytosis increases Aβ clearance, its dysregulation may be detrimental to synapse organization or for the interaction between astrocytes and the blood-brain barrier. In other terms, the duration of microglial response (i.e., transitory vs. chronic) is the distinguishing feature of microglial function [60]. An aberrant pruning process has also been observed in the development of psychiatric [61] and metabolic disorders [62]. The effects depicted so far, i.e., cell shape remodeling, migration reduction, and the stimulation of phagocytosis, are known to depend on the cytoskeletal rearrangement, which plays a crucial role in the maintenance of the cellular structure and for the determination of cell motility [63] and the process of phagocytosis [64]. The activation of migration and phagocytosis through the remodeling of the cytoskeleton is of particular importance in senescence, and therefore for the study of dystrophic and dysfunctional microglial cells [65]. It has been suggested that neurodegeneration may be secondary to aging-related microglial deterioration, which impairs the ability of cells to engage in phagocytosis and eliminate amyloid deposits in lack of changes in cytokine and chemokine secretory activity [66].

Moreover, since the reorganization of the actin cytoskeleton is under the control of several proteins, we evaluated the activity of the Rho GTPases as RhoA and Cdc42, which are involved in the regulation of actin cytoskeletal organization, migratory capacity, and phagocytosis [67]. RhoA and Cdc42 mediate Aβ-stimulated superoxide production in microglia [68] and are involved in the regulation of cell polarization [69,70]. Our results show an increase of RhoA activity in Aβ_25–35_-treated BV-2 cells, thus favoring cell migration. These data are in agreement with several research reports summarized by Aguilar et al., (2017), in which PC12 cells treated with Aβ_1-42_ showed an increase of activated RhoA and a reduction of cell survival through the inhibition of protein tyrosine phosphatase 1B (PTP1B) [71]. An increase of activated RhoA, together with a decrease of neuronal growth, are also observed in SHSY5Y cells treated with Aβ_1-40_ [72]. By reducing RhoA activation, URB597 counteracted Aβ_25–35_-induced BV-2 cell migration capacity. As reported by Diaz-Alonso et al., (2017) and Kaplan et al., (2017), RhoA expression was reduced in the brain of mice chronically treated with Δ^9^-THC, while the CB_1_ receptor knockdown in the pyramidal neurons of newborns caused RhoA protein accumulation and impaired neuronal migration [73,74].

As demonstrated by Mendoza-Naranjo et al., (2007), Aβ-induced actin remodeling and polymerization in rat hippocampal neurons is associated with an increased colocalization of both Cdc42 and Rac-1 with F-actin-rich domains [75]. The assembly and disassembly of peripheral actin filaments promote phagocytosis and localized changes in the structure of the plasma membrane [40]. Our data showed an increase of Cdc42 activity induced by URB597, which is closely related to an enhancement of BV-2 cells performing phagocytosis. These data are also in agreement with studies performed by Kurihara et al., (2006), which reported an increase of activated Cdc42 in promyelocytic HL60 cells stimulated with the CB_2_ agonist JWH015 [76].

Previous results showed that either LPS or IFN-α can induce the M1 phenotype and an increase of iNOS expression, while the M2 phenotype is induced by immunosuppressive cytokines such as TGF-α and IL-10. CB_2_ receptor activation is important for the up-regulation of Arg-1 in the M2 state and for the phagocytic function of microglia [77]. Mecha et al., (2015) showed that the stimulation with 2-AG and AEA induces an increase of Arg-1 in rat and human microglia, boosting the expression of the M2 phenotype that is reduced in CB_2_^-/-^ mice. Moreover, the pro-inflammatory LPS challenge and the transformation in the M1 phenotype are associated with the downregulation of the CB_1_, CB_2_, NAPE-PLD, FAAH, and MAGL transcripts [78]. Thus, the development of drugs able to offset or slow down the inflammatory response may add an important therapeutic option for the delay of AD progression as well as attenuate the clinical impact of AD.

Our findings support the view that the URB59-induced increase of eCB availability in the microenvironment of BV-2 microglial cells might directly interfere with the cytokine production elicited by the Aβ_25–35_ challenge. Indeed, IL-1β and TNF-α were significantly reduced just after the first hour of URB597 infusion. Previous reports showed that microglial TNF-α production after a fibrillar Aβ challenge can be inhibited by CB_2_ receptor agonist stimulation, further supporting the anti-inflammatory role attributed to the CB_2_ receptor [18,79].

Moreover, eCBs have been shown to activate anti-inflammatory signaling pathways that are able to modulate immune cell functions [80]. IL-10 was shown to increase in BV-2 one hour after the Aβ_25–35_ challenge, while after 24 h, the IL-10 gene expression dropped to the values measured in untreated cells. Similarly, an increase of IL-10 expression was observed in macrophage after 1 h in the LPS challenge, with a corresponding decrease at 24 h [81]. A reduction of IL-10 in BV-2 cells challenged with Aβ_25–35_ has also been reported in a study on the protective effect of the flavonoid genistein [82]. IL-10 production has been considered associated with disease progression; indeed, it has been found elevated in Aβ-stimulated peripheral blood mononuclear cells from patients with slow disease progression [83,84].

In our experimental conditions, URB597 infusion counteracted the Aβ_25–35_ challenge at 1 h and stimulated an increase of IL-10 production at 24h. Thus, FAAH inhibition seems to drive BV-2 polarization towards the M2 phenotype, with an increased release of the anti-inflammatory cytokines IL-10 and TGF-β. It has recently been underlined that the enhancement of eCB levels via FAAH inhibition can reduce iNOS expression in microglial cells, producing anti-inflammatory effects after stress exposure [24]. In addition, the use of PF-3845 as an irreversible FAAH inhibitor downregulated the expression of iNOS and COX-2 after traumatic brain injury [85].

In a model of ethanol-induced neurodegeneration, FAAH inhibition via a five-day URB597 treatment was shown to be effective in reducing Iba1 mRNA levels, TNF-α, IL-6, iNOS, and the monocyte chemoattractant protein-1 (MCP-1/CCL2) in hippocampal microglial cells [86]. BV-2 cells incubated with Aβ_25–35_ showed an increase of iNOS expression, reducing Arg-1 at the same time. URB597 had an opposite trend, demonstrating a counteraction of the Aβ_25–35_-induced pro-inflammatory microglial phenotype. These data are in agreement with Tham et al., (2007), which reported that URB597 treatment reduces the expression of iNOS in LPS-treated primary microglia cultures and BV-2 cells [87]. Moreover, in a mouse model of chronic cerebral hypoperfusion, Su et al., (2017) observed a reduction of the iNOS marker after URB597 treatment [88].

Altogether, our data indicate that by acting on cytoskeleton reorganization, FAAH inhibition regulates phagocytosis and cell migration processes, also promoting microglial polarization towards an anti-inflammatory M2 phenotype.

## 4. Materials and Methods

### 4.1. Materials and Chemicals

URB597 3-(3-carbamoylphenyl)-phenyl N-cyclohexylcarbamate was from Selleck Chemicals (Selleck Chemicals, Houston, TX, USA). β-amyloid peptide (Aβ_25–35_) fragment was synthesized by conventional solid phase chemistry [89]. Tissue culture medium and serum were from Gibco BRL (Life Technologies Inc., Grand Island, NY, USA). The 3-(4,5-dimethylthiazol-2-yl)-2,5 diphenyltetrazolium bromide (MTT), 4′, 6-diamidino-2-phenylindole (DAPI), and TRITC-phalloidin were purchased from Sigma-Aldrich (St. Louis, MO, USA). Fluorescein isothiocyanate (FITC)-dextran was from Sigma-Aldrich (Sigma-Aldrich, St. Louis, MO, USA). Pull-Down and Detection Kits were from Thermo Scientific Pierce, Rockford (Thermo Scientific Pierce, Rockford, IL, USA). The miRNeasy Micro kit was obtained from QIAGEN (Hilden, Germany). The High-Capacity cDNA Reverse Transcription kit and Power SYBR^®^ Green Master Mix was purchased from Applied Biosystems (Foster City, CA, USA).

### 4.2. Preparation of Aβ_25–35_ and URB597 Stock Solution

Aβ_25–35_ was dissolved in a sterile phosphate buffered saline (PBS), pH 7.4, at a concentration of 1 mM. To induce the aggregation, the solution was incubated in an ultrasonic bath on ice for 30 min. After treatment, the solution was stored at −20 °C until use. URB597 was dissolved in dimethyl sulfoxide (DMSO) at a final concentration of 1 mM.

### 4.3. Fatty Acid Amide Hydrolase Assay

FAAH enzymatic activity was studied by using membranes prepared from BV-2 cells treated with vehicle or pre-treated with 5 μM URB597 for 4 h and incubated with 30 μM Aβ25-35 for 24 h. Membrane preparations were then incubated with [^14^C]-AEA (85.0 mCi/mmol, ARC St. Louis, MO, USA) properly diluted with AEA (Cayman Chemicals, Ann Arbor, MI, USA) in 50 mM Tris-HCl, pH = 9, for 30 min at 37 °C. The [^14^C]-Ethanolamine produced from [^14^C]-AEA hydrolysis was measured by scintillation counting of the aqueous phase after the extraction of the incubation mixture, with two volumes of CHCl_3_/CH_3_OH (1:1, *v*/*v*); the activity was expressed as percentage of the maximum effect observed in the absence of treatments.

### 4.4. Cell Cultures and Treatment

The mouse microglia cell line (BV-2), kindly provided by Dr. Mangino, Sapienza University of Rome, was grown in DMEM/F-12 medium containing 5% fetal bovine serum, 4 mM L-glutamine, and 1% of penicillin-streptomycin (Gibco BRL Life Technologies Inc., Grand Island, NY, USA), at 37 °C in a humidified atmosphere with 5% CO_2._ Cells were plated at an appropriate density according to each experimental setting and treated with 30 µM Aβ_25–35_ in the presence or absence of 5 µM URB597. The cells were pre-treated with URB597 for 4 h before adding the Aβ_25–35._ The untreated cells were considered as control.

### 4.5. Cell Viability Assays

Cell viability was evaluated with MTT and Trypan blue assays. Cells were seeded in 96-well plates at a density of 3 × 10^3^/well. After cell treatments, 20 µL of a 5 mg/mL solution of MTT in PBS was added to the culture medium at a final concentration of 0.5 mg/mL and cells were incubated at 37 °C for 2 h. The supernatant was removed from each well and the formazan crystals were solubilized in 100 µL of DMSO. The optical density (OD) was measured at 570 nm, with a reference at 690 nm, using a microplate reader (Thermo Scientific Appliskan Multimode Microplate Reader). For the Trypan blue exclusion test, BV-2 were seeded onto 24-well plates at a density of 3 × 10^3^/well. After treatments, cells were detached with 1× Tripsin-EDTA, and 100 µL of the cell suspension were mixed with 100 μL of Trypan blue solution; cell counts were performed using a Neubauer chamber. The results are shown as percent viability of the treated groups relative to the control, which was considered 100%.

### 4.6. Immunocytochemical Analyses

Cells grown on 12 mm glass coverslips in a 24-well plate at a density of 15 × 10^3^ cells/well, were fixed after treatments with 4% paraformaldehyde for 30 min, followed by treatment with 0.1 M glycine in PBS for 20 min. A total of 0.1% Triton X-100 in PBS was added for 5 min to allow permeabilization. To analyze cytoskeletal actin organization, cells were labeled with rhodamine-conjugated phalloidin (TRITC-phalloidin—Sigma-Aldrich, St. Louis, MO, USA) for 45 min. For the detection of M1/M2 polarization markers, cells were incubated with primary antibodies raised against rabbit polyclonal IgG anti-Iba1 (dil. 1:100—AB-83747, Immunological Sciences, Roma, Italy), rabbit polyclonal IgG anti-iNOS (dil. 1:100—D6B65, Cell Signaling Technology, Danvers, MA, USA), or rabbit polyclonal IgG anti-Arg-1 (dil. 1:50—D4E3M, Cell Signaling Technology, Danvers, Ma, USA), and subsequently with anti-rabbit Alexa Fluor 488 secondary antibodies. Finally, the cells were marked with DAPI to highlight the nucleus. The fluorescence signal was analyzed using an Axio Observer inverted microscope, equipped with the ApoTome System (Carl Zeiss Inc., Ober Kochen, Germany). Cell area was quantified with ImageJ software.

### 4.7. Migration Assay

To evaluate the migration capacity of the cells following treatments, we performed a Scratch assay. Cells were seeded at 1 × 10^6^ cells onto 35 mm plates and grown until confluence in a DMEM/F-12 medium containing 5% fetal bovine serum. A cell-free area was introduced by scraping the monolayer with a sterile tip. After an intensive wash, the remaining cells were incubated with 30 µM Aβ_25–35_ in the presence or absence of 5 µM URB597 at different time points. After incubation, images were taken using an Axio Observer inverted microscope (Zeiss, Oberkochen, Germany). Migration was quantified by counting the cells that covered the cell-free areas by Image J, and data obtained were reported as percentage of the cells migrated in the uncovered areas with respect to time 0.

### 4.8. Fluorescein Isothiocyanate (FITC)-Dextran Uptake Assay

The phagocytic capacity was evaluated by measuring the internalization of the dextran conjugated with FITC into the cells treated with 30 µM Aβ_25–35_ in the presence or absence of 5 µM URB597. Cells were plated onto 12 mm glass coverslips in a 24-well plate at a density of 17 × 10^3^ cells/well. After treatments, the cells were incubated with dextran, conjugated with FITC in culture medium for 1 h at 37 °C, or at 4 °C for the negative control. The cells were stained with DAPI to detect the nucleus. Cells that were found positive for FITC, detected by the fluorescence signal using an Axio Observer inverted microscope (Zeiss, Oberkochen, Germany), were considered as cells that had successfully engulfed dextran. The quantitative analysis of the fluorescence intensity of the dextran and of the number of dextran-positive cells was carried out using the ImageJ software.

### 4.9. Western Blotting

Cells treated with 30 µM Aβ_25–35_ in the presence or absence of 5 µM URB597 were lysed in 100 µL/10^6^ cells of RIPA buffer (50 mM Tris-HCl pH 7.4, 150 mM NaCl, 5 mM NaF, 0.5% NaDOC, 0.1% SDS, 1 mM Na_3_VO_4_, 1 mM PMSF, 1% NP-40 and proteases inhibitors). Lysed samples were incubated on ice for 30 min and centrifuged at 14000 rpm for 15 min at 4 °C. Supernatants were collected and the protein amount was quantified using a Bradford Assay. In total, 20 µg of proteins for each sample, added with Laemmli buffer (240 mM Tris-HCl, pH 6.8, 40% glycerol, 5% β-mercaptoetanolo, 8% SDS, 0,04% di Bromophenol blue), were boiled for 3 min at 80 °C and loaded onto the 4–20% Criterion™ TGX Stain-Free™ Protein Gel (BioRad). Proteins were transferred onto PVDF membranes (BioRad) using Turbo Blot system (BioRad). Membranes were blocked with 5% Milk (*w*/*v*) in Tris-buffered saline (TBS)-Tween (0.5% (*v*/*v*)) for 1 h, and then incubated at 4 °C overnight with the rabbit polyclonal IgG anti-Iba1 (dil. 1:100 AB-81359, Immunological Sciences, Roma, Italy). After the washing steps with TBS-Tween (0.1%), membranes were incubated with Goat Anti-Mouse IgG (H+L)-HRP Conjugate (dil. 1:5000) (BioRad, Segrate Milano, Italy) for 1 h at RT. The protein band intensity was identified with Crescendo (Millipore) using ChemiDoc (MolecularImager^®^ ChemiDoc™mod. MP System—BioRad Laboratories) and densitometric analyses were performed with ImageLab software (Biorad) and normalized to the corresponding β-Actin controls.

### 4.10. Pull-Down Assay for Activated Rho GTPases

Pull-down assay was performed using the “Active Rho Pull-Down and Detection Kit” (Thermo scientific—n°16116, 16119). Briefly, cells treated with 30 μM Aβ_25–35_ in the presence or absence of 5 μM URB597 were plated at an appropriate density in order to have at least 500 μg of total protein to add to the column for each sample. The cells were scraped, collected in a tube, and gently rinsed once in ice-cold TBS, 1 mM PMSF, proteases inhibitors cocktail, and 1 mM Na_3_VO_4_. After this, 10^6^ cells were lysed in 100 mL of Lysis/Binding/Wash Buffer (25 mM Tris HCl, pH 7.2, 150 mM NaCl, 5 mM MgCl_2_, 1% NP-40, 5% glycerol, proteases inhibitors cocktail, 1 mM PMSF and 1 mM Na_3_VO_4_). Subsequently, the lysates were incubated on ice for 15 min and centrifuged at 16,000× *g* for 15 min at 4 °C. To ensure the pull-down procedure, with GTPγS and GDP regarded as positive and negative controls, respectively, were incubated with 500 μg of total protein. The samples were left at 30 °C for 15 min under constant stirring and the reaction was ended by mixing the sample with MgCl_2_ at a final concentration of 60 mM on ice. The supernatant of each sample was passed through a column and incubated with 100 μL Glutathione Resin (50% slurry containing 0.05% sodium azide) and 400 μg of GST-Rhotekin-RBD (5–6 mg/mL) at 4 °C for 1 h, with gentle rocking. The columns were washed three times with Lysis/Binding/Wash Buffer. Each washing step included an intermediate centrifuge at 6000× *g* for 10–30 s. The bound proteins were eluted with 2X SDS Sample Buffer (125 mM Tris HCl, pH 6.8, 2% glycerol, 4% sodium dodecyl sulfate (SDS) (*w*/*v*), 0.05% bromophenol blue and 5% β-mercaptoethanol). The samples were electrophoresed and analyzed by western blot as described above, with a rabbit IgG anti-Rho antibody (dil. 1:670—16116, Thermo Scientific Pierce, Rockford, IL, USA). The same procedure was performed in the Cdc42 pull-down assay using a mouse monoclonal IgG1 anti-Cdc42 antibody (dil. 1:167—16119, Thermo Scientific Pierce, Rockford, IL, USA).

### 4.11. Real-Time Quantitative PCR Analysis

Total RNA was extracted from control, BV-2 cells were treated using the miRNeasy Micro kit (Qiagen, Hilden, Germany) and quantified using NanoDrop One/One C (Thermo Fisher Scientific, Waltham, MA, USA). The cDNA was generated using the High-Capacity cDNA Reverse Transcription kit (Applied Biosystem, Foster City, CA, USA). Quantitative real-time PCR (qPCR) was performed for each sample in triplicate on an Applied Biosystems 7900HT Fast real-time PCR System (Applied Biosystem, Cheshire, UK) through the program SDS2.1.1 (Applied Biosystem, Foster City, CA, USA) using the Power SYBR^®^ Green PCR Master Mix (Applied Biosystem, Foster City, CA, USA). The primers for real-time PCR amplification were designed with the UCSC GENOME BROWSER (http://genome.cse.ucsc.edu/; University of California, Santa Cruz) (Table 1). The primer pair sequences were matched by BLASTn to the genome sequence to identify the primer locations with respect to the exons. A comparative threshold cycle (CT) method was used to analyze the real-time PCR data. The amount of target, normalized to the endogenous reference of 18S rDNA primers (ΔCT) and relative to the calibrator of untreated control (ΔΔCT), was calculated by the equation 2^−ΔΔCT^, as previously described [90].

### 4.12. Statistical Analyses

Data were expressed as the mean values ± standard deviations (SD) or mean values ± SEM from at least three independent experiments. Statistical analyses were performed using one-way analysis of variance (ANOVA) with Tukey’s post hoc test comparison (GraphPad Software Inc., San Diego, CA, USA). All results were considered statistically significant with *p* < 0.05.

## 5. Conclusions

Persistent inflammatory responses and chemotaxis of microglia/macrophages in the brain accelerate the progression of AD. Thus, the eCB system might represent a pivotal factor in counteracting neuroinflammation.

## Figures and Tables

**Figure 1 ijms-22-07711-f001:**
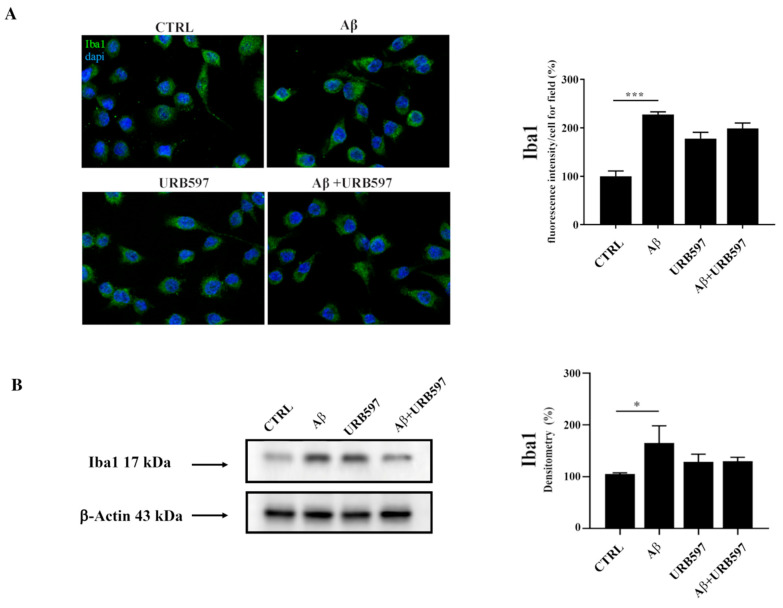
(**A**) Immunofluorescence analysis of Iba1 in BV-2 cells treated with 30 μM Aβ_25–35_ for 24 h. Quantization of the intensity of the fluorescence signal was performed using the ImageJ software. The results are expressed as the mean ± SD of three independent experiments. **** p* < 0.001. Bar 20 μm; (**B**) Determination of Iba1 protein levels by western blot. The data were normalized to the β-actin signal, reported as percentage versus control (CTRL). Data are expressed as the mean ± SEM of three independent experiments. ** p* < 0.05.

**Figure 2 ijms-22-07711-f002:**
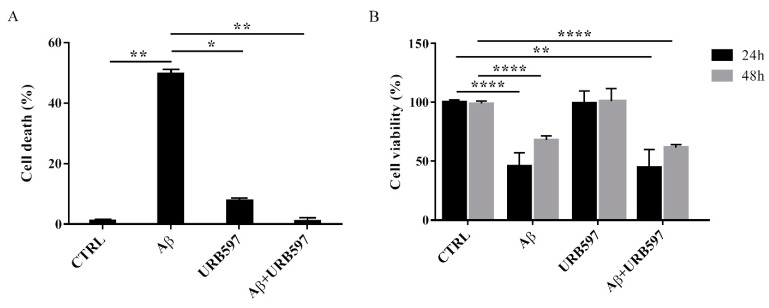
Effects of Aβ_25–35_ in the presence or absence of URB597 on BV-2 cells. (**A**) Trypan blue exclusion test. Cell count was determined in BV-2 cells exposed for 24 h to 30 µM of Aβ_25–35_ in the presence or absence of 5 µM of URB597 and expressed as cell death (cell death/cell death + living cell). Data are reported as percentage versus CTRL. (**B**) Analysis of cell viability was evaluated by MTT assay. MTT reduction was analyzed in the same samples of treated cells at 24 h and 48 h. Data are expressed as percentage versus CTRL. The values are the mean ± SEM of triplicate determination from independent experiments. ** p* < 0.05, *** p* < 0.01, ***** p* < 0.0001.

**Figure 3 ijms-22-07711-f003:**
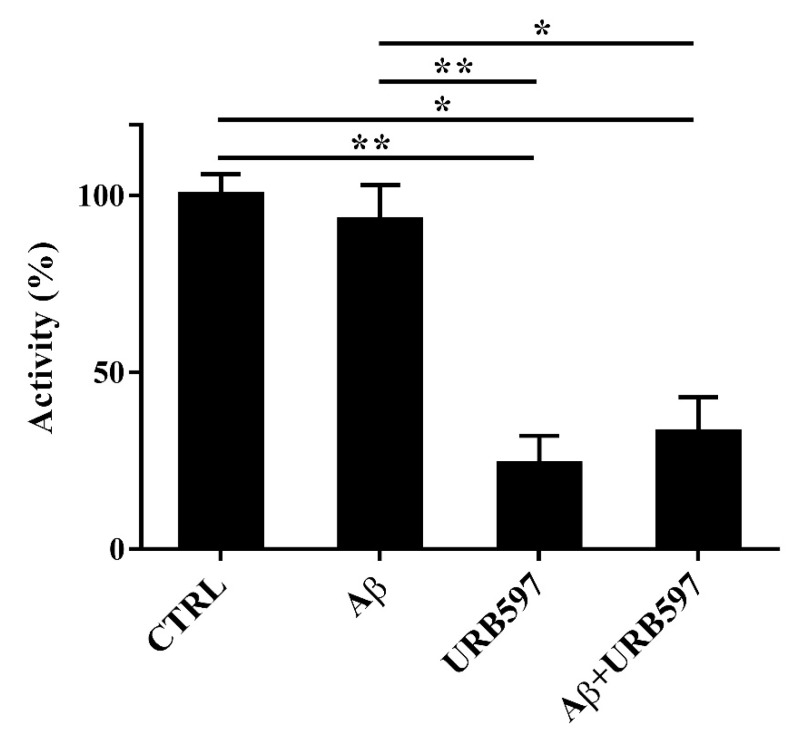
FAAH activity in BV-2 cells pre-treated with 5 μM URB597 for 4 h and incubated with 30 μM Aβ_25–35_ for 24 h. Data are reported as mean ± SEM of three independent experiments. * *p* < 0.05, ** *p* < 0.01.

**Figure 4 ijms-22-07711-f004:**
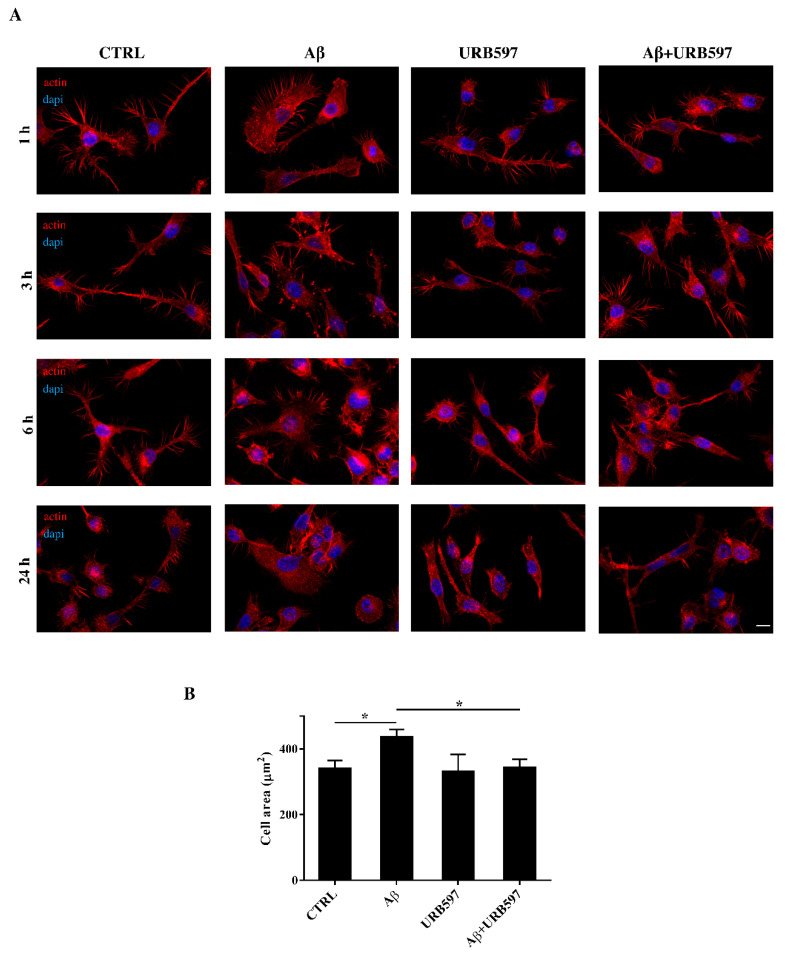
(**A**) Analysis of BV-2 cell morphology by rhodamine-conjugated phalloidin (TRITC-phalloidin) staining to highlight actin and 4′, 6-diamidino-2-phenylindole (DAPI) to detect the nucleus, after pre-treatment with 5 μM URB597 for 4 h and incubated with 30 μM Aβ_25–35_ for 1 h, 3 h, 6 h, and 24 h. (**B**) Cell areas were quantified at 24 h using Image J software. Data were reported as mean ± SD of at least three independent experiments. ** p* < 0.05. Bar: 20 µm.

**Figure 5 ijms-22-07711-f005:**
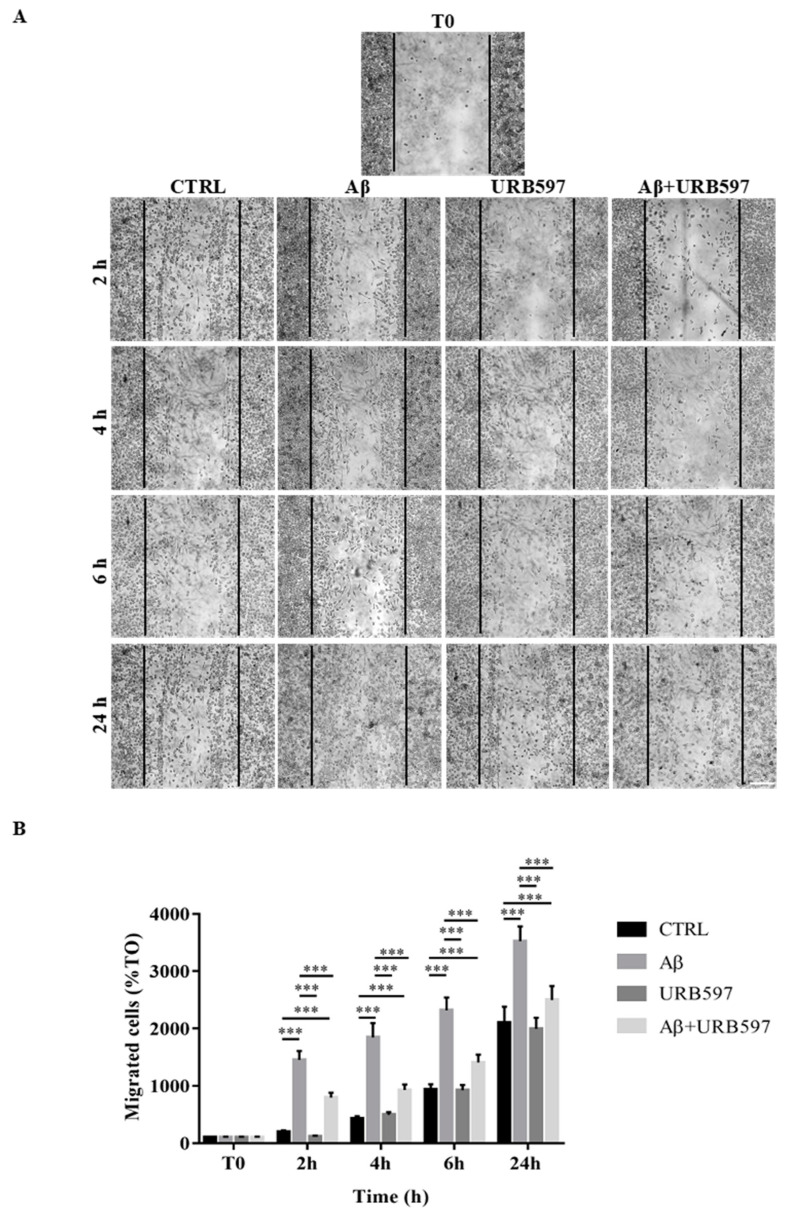
(**A**) Analysis of cell migration determined by scratch assay. BV-2 were pre-treated with 5 µM URB597 for 4 h in the presence or absence of 30 µM Aβ25–35 for 2 h, 4 h, 6 h, and 24 h. T0 represents the control at time 0, and CTRL the control for each time point. (**B**) The results were normalized vs. T0 sample and re-ported as percentage. They represent the mean ± SD of three independent experiments. *** *p* < 0.001. Bar: 200 μm.

**Figure 6 ijms-22-07711-f006:**
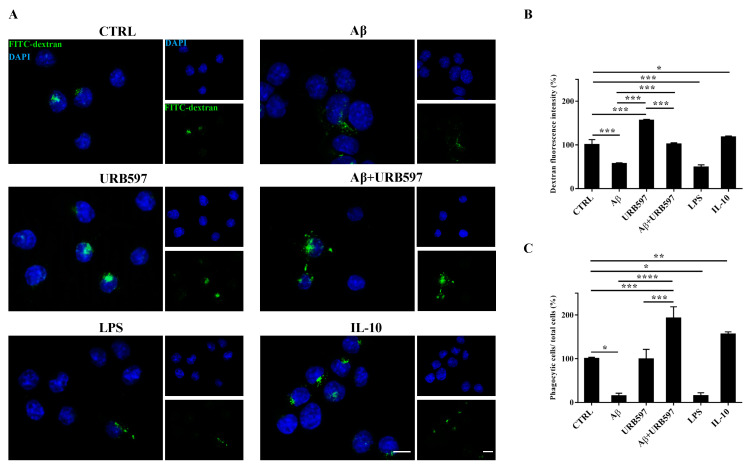
Analysis of the phagocytosis process using FITC-dextran on BV-2 cells pre-treated with 5 μM URB597 and incubated with 30 μM Aβ_25–35_ for 24 h. (**A**) Representative immunofluorescence images of FITC-dextran uptake. (**B**) Quantization of the intensity of FITC-dextran. (**C**) Quantization of the number of dextran-positive cells on the total cells. The data were normalized to CTRL and reported as a percentage. The results were expressed as the mean ± SD of three independent experiments. Statistical significance was determined using ANOVA analysis by Tukey’s test. * *p* < 0.05, ** *p* < 0.01, *** *p* < 0.001, **** *p* < 0.0001. Bar: 20 µm.

**Figure 7 ijms-22-07711-f007:**
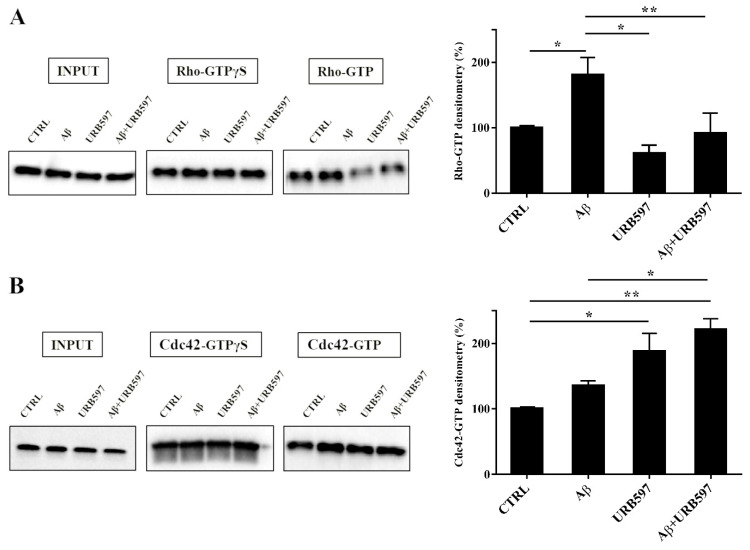
Pull-down assay of Rho and Cdc42 proteins (**A**,**B**). BV-2 cells were pre-treated with 5 μM URB597 for 4 h and incubated with 30 μM Aβ_25–35_ for 24 h. The pull-down assay with GTPγS was used as positive control. The results are reported as a percentage versus control (CTRL). Densitometric analyses were performed with ImageLab software (Biorad) and normalized to INPUT. * *p* < 0.05, ** *p* < 0.01.

**Figure 8 ijms-22-07711-f008:**
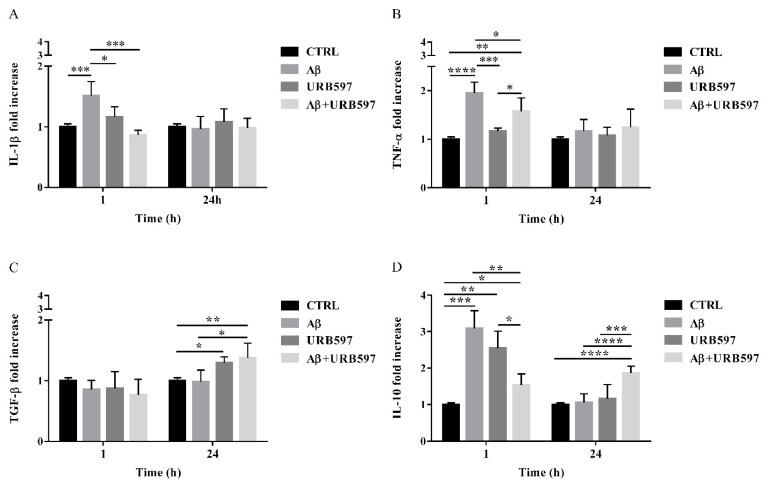
The mRNA expression of different inflammatory cytokines such as IL-1 β (**A**), TNF-α (**B**), TGF-β (**C**), and IL-10 (**D**), monitored by qPCR and normalized to 18S ribosome subunit. Data are shown as mean ± SD from three independent experiments performed in triplicate. Expression profiles were determined using the 2−ΔΔCT method. * *p* < 0.05, ** *p* < 0.01, *** *p* < 0.001, **** *p* < 0.0001.

**Figure 9 ijms-22-07711-f009:**
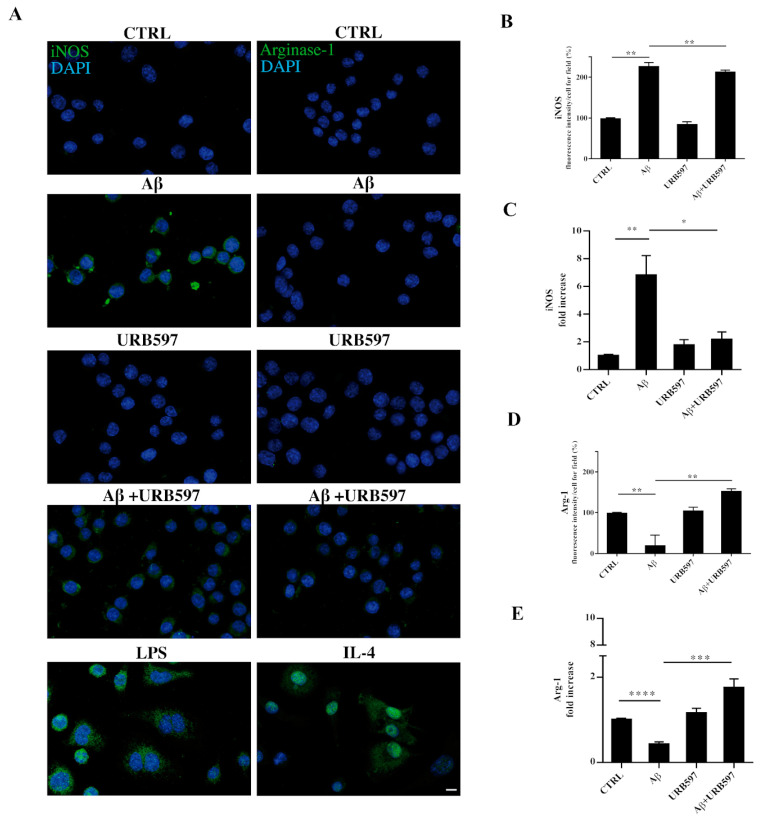
BV-2 cells were pre-treated with URB597 5μM for 4 h and incubated with Aβ_25–35_ 30 μM for 24 h. LPS and IL-4 were used as positive controls for iNOS and Arg-1 markers, respectively. Representative immunofluorescence images of iNOS and Arg-1 (**A**). Quantitative immunofluorescence analysis of iNOS (**B**) and Arg-1 (**D**). Quantification of fluorescence signal intensity was analyzed using ImageJ software. The results are expressed as the mean ± SD of three independent experiments. ** *p* < 0.01. Bar 20 μm. iNOS (**C**) and Arg-1 (**E**) mRNA expressions were evaluated by qRT-PCR at 24 h. Data are shown as mean ± SD from three independent experiments performed in triplicate. Expression profiles were determined using the 2−ΔΔCT method. * *p* < 0.05, ** *p* < 0.01. *** *p* < 0.001, **** *p* < 0.0001.

**Table 1 ijms-22-07711-t001:** List of primer couples generated for qPCR.

GENE	Forward Primer (5′–3′)	Reverse Primer (5′–3′)	Accession Numbers
mIL-1β	GAAATGCCACCTTTTGACAGTG	TGGATGCTCTCATCAGGACAG	NM_008361.4
mTNF-α	CTGAACTTCGGGGTGATCGG	GGCTTGTCACTCGAATTTTGAGA	BC137720.1
mTGF-β	CTCCCGTGGCTTCTAGTGC	GCCTTAGTTTGGACAGGATCTG	NM_011577.2
mIL-10	GCCCTTTGCTATGGTGTCCTTTC	TCCCTGGTTTCTCTTCCCAAGAC	NM_010548.2
mR18s	AAATCAGTTATGGTTCCTTTGGTC	GCTCTAGAATTACCACAGTTATCCAA	M27358
mARG1	ATGTGCCCTCTGTCTTTTAGGG	GGTCTCTCACGTCATACTCTGT	NM_007482.3
miNOS	GGCAGCCTGTGAGACCTTTG	GCATTGGAAGTGAAGCGTTTC	AF427516.1

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
