# Peer review of "Fatty Acid Amide Hydrolase (FAAH) Inhibition Modulates Amyloid-Beta-Induced Microglia Polarization"

_ijms, 2021, doi:10.3390/ijms22147711_

Round 1
Reviewer 1 Report
The paper by Grieco and coauthors analyzes how morphological dynamics of the actin cytoskeleton, migration, phagocytosis and the polarization of the microglia immortalized cell line BV2 affected by Ab peptide are restored by the FAAH inhibitor URB597. These results point out that the inhibition of the N-arachidonoylethanolamine degradation can reduce microglial activation and induce an anti-inflammatory phenotype. The study is interesting and deals with an up-to-date topic, the role of microglia in AD, and the possible strategies to drive its phenotypes toward a neuroprotective state. Yet, several issues appear weak and should be improved to obtain more compelling results.
1) Fig. 1: Iba 1 is used as a microglial activation marker. This protein is often upregulated in neurodegenerative conditions not because cells increase its expression, but just because of massive recruitment and proliferation of microglia. Thus, the conclusion that Iba1 upregulation in treated cells (and a western blot, to provide more quantitative data should be provided) indicates microglia activation is questionable. Rather, fluorescence stainings with M1-specific activation markers, such as CD68, Trem2 or CX3CR1, would be more appropriate.
2) Fig. 2: In a) the Authors demonstrate that Ab induces cell death and that URB597 protects BV2 cells. In b) they show that Ab decreases cell viability while the addition of URB597 does not promote survival. These data seem to be conflicting and an explanation of these results is lacking. I’m also quite surprised by the effect induced by Ab in these cells, as in animal models of AD and post-mortem tissues from patients, microglia displays elevated proliferation that is one of the main features of the neuroinflammatory process. How does this pro-death effect of Ab on microglia fit with what occurs in AD models and patients? The Authors should discuss this issue.
3) The Authors state that “The activation of Rho GTPases supports the hypothesis of Aβ-driven cytoskeleton rearrangement ultimately, leading to an increase of migration and phagocytosis. Since these effects were counteracted by URB597, we aimed at determining …” (lines 267-269). I’m confused, isn’t Ab inhibiting phagocytosis?
4) Fig. 10: The quantification of M1/M2 markers should be performed by RT-qPCR. In particular, the decrease of arginase induced by Ab is not evident, as the control shows negative staining for arginase.
5) Although impairment of phagocytosis is associated with AD, on the opposite the reactivation of uncontrolled phagocytosis, leading to aberrant synaptic pruning can account for neurocognitive defects in the disease (see for example Rajendran et al., 2018). This aspect should be discussed by the Authors.
Minor
1) Fig. 4B: it is not indicated to which time point the quantification is referred
2) Fig. 5: is it possible to replace the images with some of better quality or at least to reduce and uniform the background? It is quite difficult to visualize cells in several of the shown fields. Moreover, since it is reported in the methods section that “migration was quantified by measuring the recovered scratched area” why is the label of the Y-axis tagged as “wound width”? In the images presented, it would be impossible to establish the width of the scratch because cells do not form a compact border as sometimes happens in these types of assays. Moreover, how do you exclude an effect of proliferation in the recovery of the scratched area? Were treatments performed in low serum concentration? This should be added in the methods section.
3) Fig. 6: it is not clear how the bead fluorescence was detected. Since the cells were plated on a coverslip I suppose it was done by microscopy. This should be added, and maybe an image will provide a nice complement to this figure.
4) Fig. 7: I think that this Figure is dispensable since I guess that the same result is represented also in the input panels of Fig. 8 A and B, which are not described in the legend.
5) The discussion is very thorough but is maybe too long: lines 311-320 can be easily summarized in one sentence with an appropriate reference and also the paragraph related to Rho and Cdc42 engagement, as well as that of microglia M1/M2 polarization, can be shortened.
6) I suggest adding a figure showing the mechanisms affected by Ab and the pathways that are inhibited by URB597, to highlight the issue that Ab doesn’t act on FAAH but the inhibition of the enzyme can limit the pro-inflammatory function exerted by Ab, with possible connections between the two pathways.
Author Response
- 1: Iba 1 is used as a microglial activation marker. This protein is often upregulated in neurodegenerative conditions not because cells increase its expression, but just because of massive recruitment and proliferation of microglia. Thus, the conclusion that Iba1 upregulation in treated cells (and a western blot, to provide more quantitative data should be provided) indicates microglia activation is questionable. Rather, fluorescence stainings with M1-specific activation markers, such as CD68, Trem2 or CX3CR1, would be more appropriate.
We appreciate this suggestion and we have now included a western blot analysis of Iba1 expression in the result section, as well as a new panel inside figure 1. In addition, to evaluate the M1 or M2 condition in the treated cells we performed RT-PCR analysis to determine the level of nitric oxide synthase (iNOS) and Arginase-1 (Arg-1), which are markers of M1 and M2 microglia phenotypes, respectively. This is now expressed as a new panel of figure 9.
- 2: In a) the Authors demonstrate that Ab induces cell death and that URB597 protects BV2 cells. In b) they show that Ab decreases cell viability while the addition of URB597 does not promote survival. These data seem to be conflicting and an explanation of these results is lacking. I’m also quite surprised by the effect induced by Ab in these cells, as in animal models of AD and post-mortem tissues from patients, microglia displays elevated proliferation that is one of the main features of the neuroinflammatory process. How does this pro-death effect of Ab on microglia fit with what occurs in AD models and patients? The Authors should discuss this issue
We thank the referee for the question. As shown in Fig.2, although a decrease in viability of 50% is clearly present at 24 h, at 48 h there is a slightly recovery of microglia, which is reasonably due to a proliferating effect (as reported by different authors: e.g., by Di Francesco L, et al. Proteomics. 2012 Jan;12(1):124-34.). It should be also considered that the slight proliferative effect observed in our microglia cells appears less effective than that occurring in tissues and animal model probably due the different time window of observation. We should also note that the investigation of microglial proliferation goes beyond the scope of this paper; indeed, we only aimed at assessing the toxicity of URB at concentration tested without distinguishing dividing cells from non-dividing cells.
Clearly, we all agree with the reviewer about the role of increased microglia proliferation in the neuroinflammatory process, and for this reason we included a sentence in the result section highlighting the slight proliferative effect on Aβ observed at 48 h.
- The Authors state that “The activation of Rho GTPases supports the hypothesis of Aβ-driven cytoskeleton rearrangement ultimately, leading to an increase of migration and phagocytosis. Since these effects were counteracted by URB597, we aimed at determining …” (lines 267-269). I’m confused, isn’t Ab inhibiting phagocytosis?
We thank the reviewer for the observation and we amended our mistake.
- 10: The quantification of M1/M2 markers should be performed by RT-qPCR. In particular, the decrease of arginase induced by Ab is not evident, as the control shows negative staining for arginase.
We thank the reviewer for the observation, we added RT-qPCR results of iNOS and Arg-1 gene expression as required. Data are now showed as new panel in Fig. 9
5) Although impairment of phagocytosis is associated with AD, on the opposite the reactivation of uncontrolled phagocytosis, leading to aberrant synaptic pruning can account for neurocognitive defects in the disease (see for example et al., 2018). This aspect should be discussed by the Authors.
We commented on this point in the discussion section, as suggested.
Minor
- 4B: it is not indicated to which time point the quantification is referred
We thank the reviewer for the observation. The quantitative analysis was performed at 24 h. This data is now reported in the legend of Figure 4.
- 5: is it possible to replace the images with some of better quality or at least to reduce and uniform the background? It is quite difficult to visualize cells in several of the shown fields.
We reduced and make homogeneous the background, as suggested by the referee; moreover, we replaced the image of cells treated with Aβ for 6 hours, because of the too high background .
Moreover, since it is reported in the methods section that “migration was quantified by measuring the recovered scratched area” why is the label of the Y-axis tagged as “wound width”? In the images presented, it would be impossible to establish the width of the scratch because cells do not form a compact border as sometimes happens in these types of assays.
To better elucidate the results observed by microscopy images, we have performed a different quantitative analysis, which considers the number of cells that actually migrated into the scratch. This type of quantization was carried out counting cells that migrated from the border of the scratch to the uncovered areas, using Image J, and the values obtained were reported as a percentage of cells that covered the scratch with respect to the sample corresponding to time zero. So, we have replaced the graph and in the new graph the Y-axis is tagged as “migrated cells (% T0)”.
Moreover, how do you exclude an effect of proliferation in the recovery of the scratched area?
We can exclude an effect of proliferation on cell migration in the scratch area because at the time of the wound healing experiment, 2, 4, 6 and 24 hours, there was generally no proliferation of BV2 cells.
Were treatments performed in low serum concentration? This should be added in the methods section.
The treatments were carried out keeping the cells in “DMEM/7-12 medium containing 5% fetal bovine serum”, which is a serum low concentration. At any rate, we added the quoted sentence in materials and methods section.
- 6: it is not clear how the bead fluorescence was detected. Since the cells were plated on a coverslip I suppose it was done by microscopy. This should be added, and maybe an image will provide a nice complement to this figure.
We thank the reviewer for the point. We added the sentence: “….. fluorescence was detected using an Axio Observer inverted microscope (Zeiss)” in the Materials and methods section. We also provided a representative image of phagocytosis as new panel of figure 6 (Fig. 6A)
4) Fig. 7: I think that this Figure is dispensable since I guess that the same result is represented also in the input panels of Fig. 8 A and B, which are not described in the legend.
We deleted Fig. 7 as suggested.
5) The discussion is very thorough but is maybe too long: lines 311-320 can be easily summarized in one sentence with an appropriate reference and also the paragraph related to Rho and Cdc42 engagement, as well as that of microglia M1/M2 polarization, can be shortened.
We appreciated this suggestion and we modified the discussion section, accordingly.
6) I suggest adding a figure showing the mechanisms affected by Ab and the pathways that are inhibited by URB597, to highlight the issue that Ab doesn’t act on FAAH but the inhibition of the enzyme can limit the pro-inflammatory function exerted by Ab, with possible connections between the two pathways.
We appreciate this suggestion and we agree with reviewer that to provide a figure or cartoon summarizing mechanisms and potential interplays between Aβ and FAAH pathways might be useful for readers. But, in our opinion this will make discussion too speculative as based only on published data since we did not investigate in details this point. We all hope that the referee might agree with us.
Reviewer 2 Report
The present manuscript was interesting and well organized.
Some minor points of my concerns should be revised or answered.
In the present study, the novel points or the difference from the reference [27] was the stimulation compounds Aβ replacement for LPS, wasn't it?
In Figure 1 and 2, Aβ induced cell death and decreased cell viability; therefore, was Iba1 intensity quantified per cell?
In Figure 2, URB597 reverted Aβ-induced cell death but did not prevent Aβ-reduced cell viability. What was the effects of URB597? Inhibition of cell proliferation??
In Figure 4, because Aβ induced cell death and reduced cell viability, did the number of cells decrease in 24 h after Aβ? Only by the photographs, the cell number was looked similar in all groups.
Also in Figure 4, was "cell area" quantified per cell?
In the phagocytosis analysis, the authors described "Stimulation with LPS represented the negative control of phagocytosis, whereas.....the positive control". The references should be added. In my experience, LPS increased phagocytosis in BV-2 cells assessed by uptake of fluorescent microbeads. Therefore, the references why the authors used LPS as negative control should be written.
Line 215-217 "Cells treated.......to LPS-treated cells", was the sentence correct? Check it.
In Figure 6, the represented photographs should be added to understand the data easily. In addition, was A) Quantization of the intensity of FITC-dextran per well or field, not per cell?
Moreover, in Figure 6, the effects of URB597 alone and Aβ+URB597 were different between A and B. Why? What was difference between the assessments Dextran fluorescence intensity and Phagocytic cells/total cells?
Line 271 and 272, correct "interleukins" to "cytokines".
In Figure 9, the effects of URB597 on IL-10 expression at 1h, the authors wrote both Aβ and URB597 administration, alone or in combination, induced an increase of IL-10 expression. Both Aβ and URB597 alone increased IL-10 expression but Aβ+URB597 combination-increased level was lower compared to Aβ alone and to URB597 alone. Why? Did URB597 inhibit Aβ-increased IL-10 expression?
Discussion was too speculative. It should be focused and based on the present results.
Author Response
Some minor points of my concerns should be revised or answered.
In the present study, the novel points or the difference from the reference [27] was the stimulation compounds Aβ replacement for LPS, wasn't it?
Tanaka and co-workers reported on the anti-inflammatory response of FAAH inhibition in BV2 cells under LPS treatment condition by investigating the effect of two commonly used FAAH inhibitors and FAAH knockdown by siRNA in vitro. The authors contributed to shed light on the discrepancy between the pharmacological inhibition and siRNA knockdown in response to inflammation. Indeed, they highlighted that although inhibition and knockdown of FAAH have potent anti-inflammatory effects as they downregulated COX-2, iNOS, and proinflammatory cytokine expression, M2 marker expression was increased by FAAH knockdown but not by pharmacological inhibition.
In our paper, by using the same cell line, BV2 microglial cells, but a different challenge such as amyloid-β peptide we investigated how FAAH inhibitor URB597 restored actin cytoskeletal organization, migration and phagocytosis. In addition, we reported the ability of FAAH inhibitor to drive the microglia polarization towards the anti-inflammatory phenotype not only based on gene regulation but also on functional alteration, such as changes in phagocytosis.
In Figure 1 and 2, Aβ induced cell death and decreased cell viability; therefore, was Iba1 intensity quantified per cell?
The intensity of Iba1 fluorescence was quantified per cell. Moreover, in order to provide more quantitative data on Iba1 expression we have noe included western blot analysis in the result section as new panel of figure 1
In Figure 2, URB597 reverted Aβ-induced cell death but did not prevent Aβ-reduced cell viability. What was the effects of URB597? Inhibition of cell proliferation??
We thank the referee to arise the question. As shown in Figure 2, we assessed the potential toxicity of URB and its effect on metabolic activity of mitochondrial enzymes at concentration tested without distinguishing dividing cells from non-dividing cells. We cannot exclude an indirect inhibitory effect of URB on cell proliferation due to enhancement of endocannabinoid levels but further investigation is needed to better clarify this point.
In Figure 4, because Aβ induced cell death and reduced cell viability, did the number of cells decrease in 24 h after Aβ?
Within 24 hours there was a reduction in the number of cells after treatment with Aβ.
Only by the photographs, the cell number was looked similar in all groups.
We agree with the reviewer about this point. We selected the best image of cells treated with Aβ, because our aim was to highlight the cell morphology and the reorganization of the cytoskeletal actin in comparison with the other samples. For this reason, we used a magnification that allow to detect the morphological changes but not to appreciate the total number of cells.
Also in Figure 4, was "cell area" quantified per cell?
Also in Figure 4 cell area has been quantified per cell.
In the phagocytosis analysis, the authors described "Stimulation with LPS represented the negative control of phagocytosis, whereas.....the positive control". The references should be added. In my experience, LPS increased phagocytosis in BV-2 cells assessed by uptake of fluorescent microbeads. Therefore, the references why the authors used LPS as negative control should be written.
We thank the reviewer for the observation and we added references as suggested.
Line 215-217 "Cells treated.......to LPS-treated cells", was the sentence correct? Check it.
We checked and corrected the sentence.
In Figure 6, the represented photographs should be added to understand the data easily.
We provided a representative image of phagocytosis, now showed as new panel of figure 6 (Fig. 6A)
In addition, was A) Quantization of the intensity of FITC-dextran per well or field, not per cell?
Quantization of the intensity of FITC-dextran was done per field by ImageJ.
Moreover, in Figure 6, the effects of URB597 alone and Aβ+URB597 were different between A and B. Why?
Figure 6 shows that treatment with URB597 alone induces an increase in fluorescence intensity (B) and a similar number of dextran-positive cells to the control (C). In contrast, Aβ + URB597 treatment causes a significant increase in the number of dextran positive cells (C) while the fluorescence intensity was a similar to the control (B). The efficiency of phagocytosis can be assessed by measuring by the amount of internalized dextran or by the number of cells that have engulfed FITC-dextran. Our data indicate that URB597 alone or in the presence of Aβ is capable of increasing phagocytosis. URB597 could use a different phagocytosis mechanism, depending on whether it acts alone, favoring cell engulfment, or, when in the presence of Aβ, activating a greater number of cells to phagocytosis by a still unknown mechanism.
What was difference between the assessments Dextran fluorescence intensity and Phagocytic cells/total cells?
The fluorescence intensity of dextran and the ratio of the cells that have internalized the dextran to the number of total cells, are two different ways of representing the cell phagocytic ability. Specifically, the fluorescence intensity indicates the quantity of internalized material, which can represent the phagocytosis capacity. On the other hand, the number of positive cells for dextran with respect to the total number of cells, indicates how efficient the cell phagocytosis is, standing for phagocytic index.
Line 271 and 272, correct "interleukins" to "cytokines".
We modified this point, as suggested.
In Figure 9, the effects of URB597 on IL-10 expression at 1h, the authors wrote both Aβ and URB597 administration, alone or in combination, induced an increase of IL-10 expression. Both Aβ and URB597 alone increased IL-10 expression but Aβ+URB597 combination-increased level was lower compared to Aβ alone and to URB597 alone. Why? Did URB597 inhibit Aβ-increased IL-10 expression?
We cannot exclude an inhibitory or antagonistic effect of URB on Aβ-induced IL10 expression. It would be interesting to investigate the molecular mechanism that might underly this putative interaction but further investigation is needed to better clarify this point.
Discussion was too speculative. It should be focused and based on the present results.
We appreciate this suggestion and we modified the discussion section as suggested.
Round 2
Reviewer 1 Report
The Authors properly answered to all my concerns.
Author Response
We wish to thank the reviewer for approving the revised version of manuscript isms-1233185
Reviewer 2 Report
The authors sincerely responded the reviewer's comments and improved the manuscript.
Author Response
We wish to thank the reviewer for approving the revised version of manuscript ijms-1233185